# Desiccation-Tolerance of *Mycobacterium avium*, *Mycobacterium intracellulare*, *Mycobacterium chimaera*, *Mycobacterium abscessus* and *Mycobacterium chelonae*

**DOI:** 10.3390/pathogens11040463

**Published:** 2022-04-13

**Authors:** Joseph O. Falkinham, Myra D. Williams

**Affiliations:** Department of Biological Sciences, Virginia Tech, Blacksburg, VA 24061, USA; mywillia@vt.edu

**Keywords:** *Mycobacterium avium* complex (MAC), *Mycobacterium chimaera*, desiccation-tolerance, biofilms, filter paper survival, stainless-steel survival, heater-coolers

## Abstract

Desiccation-tolerance of cells of four strains of *Mycobacterium chimaera* and individual strains of *Mycobacterium avium*, *Mycobacterium intracellulare*, *Mycobacterium abscessus*, and *Mycobacterium chelonae* were measured by two methods. The survival of water-acclimated cells both in filter paper and on the surface of stainless-steel coupons were measured. In filter paper at 40% relative humidity at 25 °C, survival of patient isolates of *M. avium* and *M. chimaera* cells was 28% and 34% after 21 days of incubation, whereas it was 100% for the Sorin 3T isolate of *M. chimaera*. On stainless-steel biofilms after 42 days of incubation at 40% relative humidity at 25 °C, survival of water-acclimated cells of *M. intracellulare* was above 100%, while *M. chelonae* cells did not survive beyond 21 days, and survival of water-acclimated cells of *M. avium* and *M. abscessus* was 18% and 14%, respectively. On stainless-steel coupons, survival of patient and Sorin 3T isolates of *M. chimaera* was quite similar, specifically between 14% and 28% survival, after 42 days of incubation at 40% relative humidity at 25 °C. The experiments would support the hypothesis that some nontuberculous mycobacterial species are relatively desiccation-tolerant, whereas others are not. Further, long-term survival of the two *M. chimaera* strains is consistent with the presence of that species in Sorin 3T heater-coolers shipped throughout the world.

## 1. Introduction

Members of the *Mycobacterium avium* complex (MAC) and other nontuberculous mycobacteria (NTM) are environmental opportunistic pathogens [1]. Importantly for human infection, NTM habitats include drinking water distribution systems and premise plumbing, most notably hospitals and homes [1]. Thus, humans are surrounded by NTM, and these environmental opportunistic pathogens should demonstrate adaptations to environmental stresses, including desiccation.

Recently, *Mycobacterium chimaera* infections were reported in patients following cardiovascular surgery [2]. *M. chimaera* is a member of the *M. avium* complex (MAC) and is widely distributed in the human environment [3]. Following the reports of heater-cooler-linked infections, the U.S. Food and Drug Administration [4] and the European Centre for Disease Prevention and Control [5] issued alerts. The *M. chimaera* infections were traced to the presence of the infecting *M. chimaera* in the water reservoirs of Sorin 3T heater-coolers used to control patient and blood temperatures during cardiac surgery [2]. Based on the close relatedness of whole genome sequences from *M. chimaera* isolates from throughout the world [6] and of isolates of *M. chimaera* from the water in the Sorin 3T manufacturing factory in Munich, Germany, used to fill and test the Sorin 3T heater-coolers before shipping [7], it appears likely that the Sorin 3T heater-coolers were inoculated and colonized at the manufacturing facility in Munich, Germany, before shipping throughout the world. If that was the case, how did the *M. chimaera* cells survive during shipping?

The objective of the experiments described in this contribution was to test the hypothesis that *M. chimaera* cells adhering to the surfaces of the reservoirs, pipes, and pumps in the Sorin 3T heater-cooler introduced during factory testing survived desiccation after emptying the heater-coolers and shipment in a dry state throughout the world. NTM cells are quite hydrophobic due to the presence of a lipid-rich outer membrane [8]. Further, the hydrophobic NTM cells preferentially attach to surfaces and grow, forming thick biofilms containing high (e.g., 10,000/cm^2^) cell numbers [9]. Beyond the goal of measuring desiccation-tolerance of mycobacteria is the wider data on the survival of opportunistic pathogens in medical equipment and on surfaces in hospitals and long-term care facilities, as they have a higher proportion of at-risk individuals.

Further, to judge the desiccation-tolerance of Sorin 3T isolates of *M. chimaera*, the desiccation-tolerance of non-outbreak patient *M. chimaera* isolates, as well as other strains of other nontuberculous mycobacteria, were measured. Long-term survival of NTM is consistent with the report of desiccation-tolerance of *Mycobacterium avium* strains [10] and of long-term survival of *Mycobacterium abscessus* on coins [11].

## 2. Results

### 2.1. Desiccation Susceptibility of Mycobacterium spp. Cells in Filters

Desiccation survival was measured of water-acclimated *M. chimaera* cells added to filter paper incubated in a dry atmosphere, as described by Hugenholtz et al. [12]. The *M. chimaera* Sorin isolate, strain NC-W-2-1, was substantially more able to survive desiccation than either the *M. chimaera* patient isolate, strain MA3833, and the *M. avium* patient isolate, strain A5 (Table 1). In fact, rather than a reduction, the CFU after 7 and 14 days of incubation was higher than the initial value (Table 1). Statistical analysis of the results (Table 1) showed that there were significant differences (*p* < 0.05) between the initial (0) counts and those at 7, 14, and 21 days for strains *M. avium* A5 and *M. chimaera* MA3833. However, there was no statistical difference (*p* > 0.05) for the initial (0) and later counts (i.e., 7, 14, and 21 days) for *M. chimaera* NC-W-2-1. The high variation in the colony counts of the Sorin 3T isolates is likely due to the propensity of those strains to form aggregates and adhere to surfaces. Further, as each data point is the average of duplicate colony counts of two different pieces of inoculated filter paper, any variation in inoculum and filter paper moisture content would have likely influenced the values.

### 2.2. Desiccation Susceptibility of Mycobacterium spp. Cells on Stainless-Steel Coupons

To mimic conditions of *M. chimaera* survival in a heater-cooler, the survival of water-acclimated cells adhering to stainless-steel coupons was measured. As for the measurements of survival in filter paper, the *Mycobacterium* spp. cells were water-acclimated to provide cells mimicking those in drinking water. For these experiments, stainless-steel coupons were exposed to a suspension of *Mycobacterium* spp. cells in a CDC Bioreactor for 6 h. The exposure duration was chosen based on the length of time required for shipping heater-coolers from Munich, Germany, to a U.S. hospital in Pennsylvania, namely 22 days. The heater-cooler had been filled with *M. chimaera*-containing factory water for testing, prior to draining and shipping. However, in spite of draining, water containing *M. chimaera* was still in the instrument and cells had adhered to the interior surfaces of the water circuit [9]. The results (Table 2) show that the two representative strains of slowly growing *Mycobacterium*, *M. avium* and *M. intracellulare*, were more desiccation-tolerant compared to the two strains of rapidly growing *Mycobacterium*, *M. abscessus* and *Mycobacterium chelonae*. The desiccation-tolerance of the *M. intracellulare* strain was the highest. Survival of the *M. chelonae* strain was lost after 21 days of incubation. All strains listed in Table 2 had been isolated from infected patients and were not associated with cardiac surgery where a Sorin 3T had been used. Again, as noted for the measurements of surviving colony count in filter paper, the variation of counts is high, again attributable to the hydrophobicity-driven aggregation and surface adherence of the mycobacterial cells. 

### 2.3. Desiccation Tolerance of M. chimaera in Biofilms on Stainless-Steel Coupons

As shown for the slowly growing *Mycobacterium avium* complex strains, *M. avium* and *M. intracellulare*, the desiccation-tolerance of the four *M. chimaera* strains was relatively high (Table 3). Measurement of two Sorin 3T and two patient isolates of *M. chimaera* were performed to determine whether the Sorin 3T isolates were uniquely more desiccation-tolerant than the patient isolates. That was not the case (Table 3). Included with the desiccation survival data are the colony counts of the coupons measured immediately after the 6 h adherence period and after washing. It can be seen that a substantial number of water-acclimated *M. chimaera* cells adhered to the coupons within the 6 h period. Again, as noted for the measurements of colony counts in filter paper, the data show appreciable variation. However, the points represent duplicate colony counts of suspensions from individual stainless-steel coupons, where both the initial number of CFU would vary as would the stainless-steel surfaces. Those variations likely contributed to the size of the standard deviations (Table 2 and Table 3).

## 3. Discussion

Here, we have shown that cells of representative slowly growing mycobacteria, namely members of the *Mycobacterium avium* complex (MAC), are relatively desiccation-tolerant. This is consistent with their natural habitat, namely rivers, streams, lakes, and estuaries. It would be expected that one adaptation to persistence in natural waters would be the ability to survive periods of desiccation during periods of low water flow. The *M. avium* complex (MAC) strains tested here were more desiccation-tolerant than the two rapidly growing strains, *M. abscessus* and *M. chelonae* (Table 2). If, upon further investigation (see below), that is the case, a fruitful research objective would be to identify the basis for that desiccation susceptibility. Perhaps, the shorter chain lipids in rapidly growing mycobacteria do not protect against desiccation as well as the long-chain lipids of slowly growing mycobacteria [8]. However, the two strains chosen might not be representative of other strains or the growth conditions were not optimal. For example, one Reviewer pointed out that the optimal growth temperature of *M. abscessus* is 28 °C and all cultures were incubated at 37 °C. Growth and incubation at a sub-optimal growth temperature might predispose cells to increased desiccation susceptibility.

As in all studies, there are limitations to the interpretation of this study. First is the ever-present challenge of measuring colony counts and following the growth of the aggregating mycobacterial cells. The inclusion of surfactants/detergents in cultures or suspensions is discouraged because of the effects of detergents on the physiological responses of mycobacteria, namely increased susceptibility. We have relied on physical methods, primarily sonication and pelleting aggregates, to prepare cultures of suspensions whose turbidity or counts can be measured with improved replication of values. A second limitation of this study involved the growth of the *M. abscessus* strain at above its optimal temperature, which might have produced cells that were more stress-sensitive. Third, the strains chosen might not be representative of the individual species. Now that an initial picture of mycobacterial desiccation-tolerance is available, studies can be duplicated with a wider range of species and types. As an example, no isolates of *M. chimaera* from patients infected from a Sorin 3T heater-cooler were included. Fourth, we chose to measure survival on stainless-steel coupons to mimic, in part, the exposure in a Sorin 3T heater-cooler. However, there are many other habitats where the environmental, opportunistic pathogenic *Mycobacterium* spp. could adhere and form biofilms, such as different pipe surfaces like plastic or copper, or natural surfaces such as rocks.

Due to differences in growth conditions, cell preparation, and desiccation-tolerance measurement protocols, it is impossible to compare *Mycobacterium* spp. desiccation-tolerance with reports of other bacteria. Earlier, it was shown that desiccation-tolerance in *Mycobacterium smegmatis* was associated with trehalose accumulation [13], and *Mycobacterium phlei* and *M. smegmatis* cells frozen in dextran or glycerol were relatively desiccation-tolerant, surviving as long as four years [14]. However, in that work, desiccation-tolerance was likely increased by the cell’s suspension in glycerol or dextran [14]. A variety of other bacterial genera have been shown to be able to survive desiccation, for example, *Acinetobacter baumannii* [15] and *Burkholderia cepacia* [16]. Evidence that *Mycobacterium* spp. and two other opportunistic premise plumbing pathogens (OPPPs), *Burkholderia* spp. and *Acinetobacter* spp., are desiccation-tolerant [10,11,13,14,15], suggests that it may be a common characteristic of waterborne opportunistic pathogens. Desiccation-tolerance is clearly of survival benefit for waterborne microorganisms that can be exposed to periods of desiccation in streams, rivers, ponds, and lakes, and in water distribution systems and premise plumbing.

The desiccation-tolerance of the *Mycobacterium* spp. cells adhering to the surface of stainless-steel coupons is likely due to the fact that biofilms are rich in water, perhaps as high as 90%, due to the presence of the extracellular polymeric substances [17]. Here, the cells of the four *Mycobacterium* spp. (Table 2) and the four *M. chimaera* strains (Table 3) were not treated in any way to enhance survival. However, it is possible that water-acclimation performed for these measurements contributed to some increase in desiccation-tolerance, compared to a situation where cells were subjected to desiccation immediately after growth in laboratory medium. For example, desiccation might lead to the production of trehalose and concomitant desiccation-tolerance [13,15].

Finally, the data have direct application to understanding the persistence of *M. chimaera* in Sorin 3T heater-coolers following their introduction during final testing before shipping [7]. *M. chimaera* was introduced into the Sorin 3T heater-coolers at their site of manufacture in Munich, Germany [7]. Isolates of *M. chimaera* recovered from the factory water shared high sequence similarity with isolates from Sorin 3T heater-coolers and patients from around the world [6]. Functional testing of the Sorin 3T heater-coolers before shipping from the factory in Munich likely led to *M. chimaera* colonization of the instruments. Further, because of the speed of adherence of mycobacteria to surfaces and ensuing biofilm formation [8,9,18], the *M. chimaera* cells were encased in a water-rich biofilm on the interior plumbing surfaces of the Sorin 3T heater-coolers. Assuming a 3-week duration of shipping from Munich, Germany, to the eastern United States, between 7% and 39% of the cells in the initial inoculum would be present (Table 3). If the Sorin 3T heater-coolers were filled with water upon receipt, the surviving *M. chimaera* cells in the biofilm could be released to inoculate the water.

## 4. Materials and Methods

### 4.1. Mycobacterium spp. Strains

Four independently isolated *M. chimaera* strains were employed in the study: two from Sorin heater-coolers and two from patients. The strains include: *M. chimaera* strains NC-W-2-1 and SF-W-4-1, recovered from two different Sorin 3T heater-coolers, and *M. chimaera* strains MA3833 and P32-P-1, recovered from two different patients with pulmonary infections [19]. In addition to the four *M. chimaera* strains, a plasmid-free patient-isolate of *M. avium*, strain A5 [20], the type strain of *M. intracellulare*, strain TMC 1406^T^, a patient-isolate of *M. abscessus*, strain AAy-P-1, and a patient isolate of *M. chelonae*, strain EO-P-1, were employed for the measurement for desiccation-tolerance.

### 4.2. Growth of Mycobacterium spp. Strains 

All *Mycobacterium* spp. strains were grown to mid-exponential phase from a single isolated colony in Middlebrook 7H9 broth (BD, Sparks, MD, USA) containing 0.5% (v/v) glycerol and 10% (vol/vol) oleic acid albumin with aeration (60 rpm). Growth rates were measured as turbidity (absorbance at 540 nm) in 50 mL of M7H9 broth in 500 mL Nephalometry flasks (Bellco Glass, Inc., Vineland, NJ, USA) inoculated with 1 mL of a 7-day, 2 mL M7H9 broth culture grown at 37 °C with aeration (120 rpm), and cells were harvested when the culture reached mid-log phase. A loopful of each culture was streaked on M7H10 agar (Becton Dickinson, Sparks, MD, USA) containing 0.5% (vol/vol) glycerol and 10% (vol/vol) oleic acid albumin and incubated at 37 °C for 10–14 days to confirm purity and colonial type.

### 4.3. Medium for Enumeration of Mycobacterium spp. Strains

Colony formation of *Mycobacterium* spp. cells was measured on Middlebrook 7H10 agar medium containing 0.5% (wt./vol) glycerol and 10% (wt./vol) oleic albumin (M7H10, Becton Dickenson, Sparks, MD, USA). 

### 4.4. Water-Acclimation of Mycobacterium spp. Cells 

Following growth in M7H9 broth, the cells were collected by centrifugation (5000× *g* for 20 min), the supernatant medium was discarded and autoclaved, and the cell pellet was suspended in 50 mL of sterile Blacksburg tap water in a sterile 250 mL flask, and the turbidity of each suspension was adjusted to equal a Number 1 McFarland Standard. The cells were acclimated to tap water by incubation of the McFarland-adjusted cell suspensions at their growth temperature (i.e., 37 °C) for 7 days with aeration (120 rpm). To reduce the proportion of aggregated cells, the water-acclimated suspensions were subjected to low-speed centrifugation (2000× *g* for 10 min) to pellet aggregates. The resulting supernatant suspension—a reduced aggregate fraction [21]—was employed as the inoculum for the desiccation studies.

### 4.5. Measurement of Desiccation Susceptibility in Filters 

Desiccation susceptibility of the *Mycobacterium* spp. cells in filters was measured as described by Hugenholtz et al. [12]. Cells in 5 mL of each water-acclimated suspension were collected by filtration through 0.45 µm-pore size sterile filters. Immediately, the CFU/filter (control) was measured by transferring one filter into 5 mL of sterile Blacksburg tap water and vortexing at the highest speed for 1 min. After vortexing, a 10-fold dilution series in sterile Blacksburg tap water was prepared and 0.1 mL of each suspension was spread on M7H10 agar plates in triplicate. The plates were incubated at 37 °C for 14–21 days and colonies were counted to calculate CFU/mL and CFU/filter. The remaining filters were transferred to a plastic container with a cup of desiccant and incubated at room temperature. After 7, 14, and 21 days of incubation at a relative humidity of 40%, one filter was withdrawn, CFU/filter was measured, and survival was calculated.

### 4.6. Measurement of Desiccation Susceptibility in Biofilms on Stainless-Steel Coupons

To duplicate conditions of *M. chimaera* in heater-coolers, desiccation susceptibility was measured on coupons of stainless-steel. Desiccation susceptibility was measured on *Mycobacterium* spp. cells adhering to stainless-steel coupons from a CDC Bioreactor (BioSurface Technologies, Bozeman, MT, USA). Following growth and water acclimation, cells in each water-acclimated suspension were added to 350 mL of sterile Blacksburg tap water in the reservoir of a sterile CDC Bioreactor containing sterile stainless-steel coupons to a final density of 100,000 CFU/mL. After 6 h of incubation at room temperature, the paddles and coupons (3 coupons/paddle) were removed from the bioreactor and gently rinsed by emersion in 1 L of sterile Blacksburg tap water. Immediately, three coupons from one paddle were removed aseptically and the CFU/coupon was measured as described for adherence (above). The remaining rinsed and air-dried paddles with coupons were placed in a desiccation chamber and incubated at room temperature at a relative humidity of 40%. After 7, 14, 21, 28, 35, and 42 days, a paddle was removed, the coupons were separated aseptically, the CFU per paddle was measured as described above, and survival was calculated.

### 4.7. Statistical Analysis

Differences in measurements for the *Mycobacterium* spp. strains were assessed statistically using Student’s *t*-test of matched populations using GraphPad Number Software (Version 3.0, GraphPad Software, San Diego, CA, USA).

## Figures and Tables

**Table 1 pathogens-11-00463-t001:** Desiccation tolerance on filter paper of *Mycobacterium* spp. strains.

Duration Days	Percent Survival under Desiccation ^a^		
	*M. avium* A5 Patient	*M. chimaera* NC-W-2-1 Sorin 3T	*M. chimaera* MA3833 Patient
0 ^b^	100	100	100
7	28 ± 3	112 ± 20	86 ± 13
14	27 ± 0.2	150 ± 49	35 ± 5
21	28 ± 0.2	100 ± 30	34 ± 4

^a^ Average of two values for two independent measurements ± standard deviation. ^b^ CFU/filter: *M. avium* A5 = 1.5 × 10^6^, *M. chimaera* NC-W-2-1 = 1.4 × 10^6^, *M. chimaera* MA3833 = 6.2 × 10^5^.

**Table 2 pathogens-11-00463-t002:** Desiccation tolerance of *Mycobacterium* spp. in biofilms on stainless-steel coupons.

Duration Days	Percent Survival under Desiccation ^a^			
	*M. avium*	*M. intracellulare*	*M. abscessus*	*M. chelonae*
0 ^b^	100	100	100	100
7	119 ± 8	105 ± 22	72 ± 15	25 ± 16
14	69 ± 24	61 ± 4	118 ± 11	14 ± 5
21	46 ± 4	73 ± 11	10 ± 6	4 ± 3
35	42 ± 32	102 ± 9	0.6 ± 0.5	<0.02
42	18 ± 10	119 ± 5	14 ± 4	<0.04

^a^ Average of two values for two independent measurements ± standard deviation. ^b^ CFU/cm^2^: *M. avium* = 1.0 × 10^5^, *M. intracellulare* = 7.5 × 10^4^, *M abscessus* = 4.3 × 10^4^, *M. chelonae* 5.4 × 10^4^.

**Table 3 pathogens-11-00463-t003:** Desiccation susceptibility of *Mycobacterium chimaera* strains in biofilms on stainless-steel coupons.

Days	Percent Survival under Desiccation ^a^			
	NC-W-2-1 Sorin 3T	SF-W-4-1 Sorin 3T	MA3833 Patient	P32-P-1 Patient
0 ^b^	100	100	100	100
7	94 ± 25	59 ± 38	125 ± 86	84 ± 67
14	88 ± 47	23 ± 13	47 ± 15	40 ± 6
21	39 ± 18	7 ± 3	24 ± 23	31 ± 15
28	30 ± 6	17 ± 12	29 ± 5	35 ± 12
42	15 ± 4	14 ± 13	28 ± 4	19 ± 0.7

^a^ Average of two values for two independent measurements ± standard deviation. ^b^ CFU/cm^2^: Strain NC-W-2-1 = 2.3 × 10^5^, SF-W-4-1 = 5.4 × 10^5^, MA3833 = 9.7 × 10^4^, P32-P-1 = 1.15 × 10^5^.

## Data Availability

The primary data are available in the Falkinham laboratory at Virginia Tech.

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
