# Peer review of "Desiccation-Tolerance of Mycobacterium avium, Mycobacterium intracellulare, Mycobacterium chimaera, Mycobacterium abscessus and Mycobacterium chelonae"

_pathogens, 2022, doi:10.3390/pathogens11040463_

Round 1

Reviewer 1 Report

In this paper, the researchers examine the survival of various species of mycobacteria in dry conditions, both on membranes and on stainless steel coupons. The survival of fast-growing mycobacteria such as M. abscessus and M. chelonae and slow-growing mycobacteria from the MAC group with emphasis on M. chimaera was examined.

The researchers found that fast-growing mycobacteria survive less well under these conditions. Beyond that, they demonstrate the ability of M. chimaera to survive long (at least 42 days) under these desiccation conditions. These result could explain the spread of the M. chimaera pandemic strain connected to Sorin 3T heater-coolers units in the world. In a short literature review, I could not find any other similar publications on M. chimaera and therefore, there is no doubt that this article is important to readers and can improve our understanding of the M. chimaera global outbreak. Beyond that, NTMs are important emerging pathogens and recognizing their ability to survive on surfaces and / or other medical equipment may enhance the preventive measures we can take against them.

Specific comments appear as stickers in the body of the article.

Author Response

Response to Reviewer 1

Comment. We appreciate the use of “sticker” in the body of the text to suggest changes. Those changes are listed below in order of their presentation as well as our revisions of the manuscript.

  1. Table 1. As requested, the text describing the results shown in Table 1 now includes a summary of the statistical analysis. It is likely that the variation in the counts due to aggregate formation by mycobacteria, contributed to the lack of statistical significance.
  2. Tables 1, 2, and 3. We have added the CFU/filter (Table 1) or CFU/cm3 (Tables 2 and 3) as Footnotes to each Table.
  3. Lines 89-91. We have added information on the duration of shipping a new Sorin 3T heater-cooler from the factory in Munich, Germany to the York Hospital in York, Pennsylvania; namely 22 days.
  4. The Discussion has been revised as follows: (1) An additional initial paragraph lists the main findings of the study, (2) We now offer a speculation as to why the desiccation tolerance of the slowly growing Mycobacterium spp. strains was greater than that of the rapidly growing Mycobacterium spp. strains. Primarily, we suggest that the thicker outer membrane with its longer chain fatty acids of the slowly growing Mycobacterium spp. strains offer greater retention of water and thereby survival, (3) We have expanded on the possible limitations, including those noted by Reviewer 2 (i.e., growth temperature of M. abscessus not at its optimum and the absence of M. chimaera strains from patients whose infection was linked to Sorin 3T heater-coolers to compare desiccation-survival.

Reviewer 2 Report

The manuscript reports the desiccation-tolerance of cells of NTM clinical isolates and waterborne NTM isolates. The later were two Mycobacterium chimaera isolates from two different heater-cooler units used during cardiac surgeries. The desiccation-tolerance of strains cells was measured by two methods, on paper filter and in biofilms on stainless steel coupons, and the results were compared. The results showed that some nontuberculous mycobacterial species are more desiccation-tolerant then others depending on the conditions of growth and maybe the predisposition to form cell aggregates and biofilm.

This is an important work contributing to understand NTM resilience and adaptations to stresses in anthropogenic environments and, most of all, to underline the necessity of making sure that the development of accurate protocols and appropriate measures are taken to ensure the microbiological safety of waters and equipment in order to protect the population…. although the efforts (Schreiber, et al 2016 – “Reemergence of Mycobacterium chimaera in Heater-Cooler Units despite Intensified Cleaning and Disinfection Protocol”). Moreover, with the incidence of chronic diseases, human populations are becoming more susceptible to opportunistic infections. These concerns should be emphasized in the manuscript, so once and for all, accurate and rigorous strategies for water quality control start being implemented urgently.

The objective of the work was accomplished. Conclusions correspond to findings. Appropriate controls were performed. An accurate comparison of the obtained results with the previous literature has been performed. Therefore, the publication is recommended, but after some revisions and description of necessary details due to the following issues/questions:

- M&M, lines 190-192: a reference should be added indicating the previous origin of patients´ isolates: M. avium strain A5, M. abscessus strain AAy-P-1, and M. chelonae strain EO-P-1. Also the method (conserved genes sequencing, whole genome sequencing, DNA fingerprinting, etc…) used on the identification of isolates should be indicated in the text. Please, clarify the same for M. chimaera isolates obtained from patients. In case of mycobacteria, the 16SRNA gene sequencing is not sufficient to distinguish to species level (e.g. Devulder et al, 2005).

- Who works with NTM knows the difficulty on growing these strains and measuring turbidity due to cells aggregates. Could the authors explain by indicating in M&M the strategies used to circumvent cells aggregation, so others could reproduce the growth of the strains? The concentration of glycerol used doesn´t seems sufficient in order to achieve the homogeneity of the suspensions, particularly in the case of M. avium and M. abscessus strains, which cultures are typically very dry and form large cells aggregates in liquid media. Were you able to get to suspensions adjusted to equal a Number 1 McFarland Standard without cells aggregates?

- Did the authors had in consideration the optimal temperature of growth for M. abscessus, which is 28ºC or did you performed the growth of all strains at 37ºC. If not, please indicate the exception for M. abscessus. If yes, could the authors comment how it might have or not affected the results and the CFUs counting?

- What in my opinion can be a fragile point in this work was the fact that the clinical isolates of M. chimaera were obtained from patients with respiratory infections from a previous study and not from patients that had contact with the heater-coolers during cardiopulmonary surgeries, in order to obtain a more accurate comparative study.  Please, discuss/comment on this matter on the Discussion section.

- Do the authors have data on the desiccation tolerance of the M. chimaera isolates SF-W-4-1 Sorin 3T and P32-P-1 patient, on Filter Paper? There are no results, neither in the tables, nor in the text. If yes, please add data to the tables.

Author Response

Response to Reviewer 2

  1. Introduction, Lines 54-58. Additional information has been added to the Introduction to underscore the importance of infections caused by opportunistic pathogens. The sentence added is: Beyond the goal of measuring desiccation-tolerance of mycobacteria is the wider data on survival of opportunistic pathogens in medical equipment and on surfaces in hospitals and long-term care facilities, as they have a higher proportion of at-risk individuals.
  2. Materials and Methods, Lines 186-192. More information concerning the origin/source of each of the strains has been listed along with references. The revised section on Mycobacterium Strains: The strains include: M. chimaera strains NC-W-2-1 and SF-W-4-1, recovered from two different Sorin 3T heater-coolers and M. chimaera strains MA3833 and P32-P-1, recovered from two different patients with pulmonary infections [19]. In addition to the four M. chimaera strains, a plasmid-free patient-isolate of M. avium, strain A5 [20], the type strain of M. intracellulare, strain TMC 1406T, a patient-isolate of M. abscessus, strain AAy-P-1, and a patient isolate of M. chelonae, strain EO-P-1 were employed for measurement for desiccation-tolerance. Note that a citation for M. avium A5 strain has been added [20].
  3. Materials and Methods, Lines 208-213. Upon reading this Reviewer’s question concerning aggregation of mycobacterial cells, we realized we inadvertently omitted a method that is our standard lab practice from the submitted manuscript, namely the preparation of a “Reduced Aggregate Fraction” (RAF). The RAF is prepared by collecting the suspended supernatant cells from a low-speed centrifugation of the water-acclimated cell suspensions to pellet the large (visible) aggregates. A reference for the method and its employment has been added [21]. Although treatments such as growth in detergent produce smooth, aggregate-free cultures, detergent-grown cells are substantially more susceptible to disinfection and desiccation. In our experience, glycerol (employed as carbon source), does not alter aggregation.
  4. Limitations of the Study. The following limitations have been included in a paragraph on limitations of the study:
    1. The Reviewer is correct that we did not grow abscessus at its optimal temperature for growth. It might have impacted on survival, as slower growth rates of Mycobacterium spp. strains increased disinfectant survival (Taylor et al., 2000), likely permitting repair of stress-induced damage before cell division.
    2. The Reviewer is correct that the infections caused by the “comparison” chimaera patient strains were not associated with exposure to Sorin 3T heater-coolers.
    3. Limitations of the Study. We did not measure the desiccation survival of strains SF-W-4-1 or P32-P-1 in filter paper. We feel that filter paper desiccation survival is not particularly relevant to these mycobacteria that travel, attach, and grow in premise plumbing.

Round 2

Reviewer 2 Report

The persistence of NTM in waters and surfaces, associated with their high resistance to heat and to other disinfection strategies, calls for the implementation of measures to reduce human exposure to these potential pathogens, and because of that I consider this work in line with this problem and important to address some issues.

All questions and comments addressed were taken into consideration. After the revision, my recommendation is the acceptance of the manuscript in the present form.

Thank you